# Blockade of Melatonin Receptors Abolishes Its Antiarrhythmic Effect and Slows Ventricular Conduction in Rat Hearts

**DOI:** 10.3390/ijms241511931

**Published:** 2023-07-25

**Authors:** Aleksandra V. Durkina, Barbara Szeiffova Bacova, Olesya G. Bernikova, Mikhail A. Gonotkov, Ksenia A. Sedova, Julie Cuprova, Marina A. Vaykshnorayte, Emiliano R. Diez, Natalia J. Prado, Jan E. Azarov

**Affiliations:** 1Department of Cardiac Physiology, Institute of Physiology, Komi Science Center, Ural Branch of the Russian Academy of Sciences, 167982 Syktyvkar, Russia; sashadurkina@mail.ru (A.V.D.); bernikovaog@gmail.com (O.G.B.); suomi21@list.ru (M.A.G.); m.vaykshnorayte@mail.ru (M.A.V.); j.azarov@gmail.com (J.E.A.); 2Center of Experimental Medicine, Institute for Heart Research, Slovak Academy of Sciences, 81438 Bratislava, Slovakia; 3Department of Biomedical Technology, Faculty of Biomedical Engineering, Czech Technical University in Prague, 27201 Kladno, Czech Republic; sedova.ks@gmail.com; 4Department of Health Care Disciplines and Population Protection, Faculty of Biomedical Engineering, Czech Technical University in Prague, 27201 Kladno, Czech Republic; efremyul@fbmi.cvut.cz; 5Instituto de Fisiología, Facultad de Ciencias Médicas, Universidad Nacional de Cuyo, Mendoza 5500, Argentina; emiradiez@gmail.com (E.R.D.); natyprado@gmail.com (N.J.P.)

**Keywords:** rat heart, post-ischemic arrhythmias, melatonin, conduction velocity, potassium current, sodium current, connexin-43

## Abstract

Melatonin has been reported to cause myocardial electrophysiological changes and prevent ventricular tachycardia or fibrillation (VT/VF) in ischemia and reperfusion. We sought to identify electrophysiological targets responsible for the melatonin antiarrhythmic action and to explore whether melatonin receptor-dependent pathways or its antioxidative properties are essential for these effects. Ischemia was induced in anesthetized rats given a placebo, melatonin, and/or luzindole (MT1/MT2 melatonin receptor blocker), and epicardial mapping with reperfusion VT/VFs assessment was performed. The oxidative stress assessment and Western blotting analysis were performed in the explanted hearts. Transmembrane potentials and ionic currents were recorded in cardiomyocytes with melatonin and/or luzindole application. Melatonin reduced reperfusion VT/VF incidence associated with local activation time in logistic regression analysis. Melatonin prevented ischemia-related conduction slowing and did not change the total connexin43 (Cx43) level or oxidative stress markers, but it increased the content of a phosphorylated Cx43 variant (P-Cx43^368^). Luzindole abolished the melatonin antiarrhythmic effect, slowed conduction, decreased total Cx43, protein kinase Cε and P-Cx43^368^ levels, and the IK1 current, and caused resting membrane potential (RMP) depolarization. Neither melatonin nor luzindole modified INa current. Thus, the antiarrhythmic effect of melatonin was mediated by the receptor-dependent enhancement of impulse conduction, which was associated with Cx43 phosphorylation and maintaining the RMP level.

## 1. Introduction

Melatonin, a secretory product of the pineal gland, serves several physiological functions, and the regulation of circadian rhythms is probably the best-known one. However, recent studies have demonstrated other roles that melatonin can play in normal conditions and disease in many organs and tissues. Due to its amphiphilic properties, melatonin can readily enter the cell and mitochondria, where it has been reported to confer organ protection due to its versatile antioxidative effects [1,2] mediated by both direct reactive oxygen species (ROS)-scavenging [3] and stimulation of antioxidative enzymatic activity [4,5,6]. On the other hand, melatonin also interacts with multiple targets, including plasma membrane receptors, nuclear receptors (probably), and intracellular and even extracellular proteins [7].

MT1 and/or MT2 plasma membrane receptors of melatonin are members of class A of the G-protein-coupled receptor (GPCR) family. These receptors are thoroughly characterized from structural [8] and functional [9] perspectives. These receptors interact mainly with Gi/o and Gq/11 protein subfamilies [10] and involve complex intracellular signaling including, but not limited to, protein kinase A and phospholipase C-dependent pathways [9,11]. Since melatonin can enter the nucleus, the existence of nuclear melatonin receptors has been suggested. However, the identification of such receptors appears challenging. Retinoic acid-related orphan receptors (ROR) were considered as promising candidates, but findings concerning their role are controversial [7,12,13]. There are reasons to believe that ROR activation by melatonin is indirect. Recently, vitamin D receptors (VDR) have been reported to bind melatonin [14] and thereby might function as the nuclear melatonin target, which warrants further exploration of the VDR role in melatonin signaling. Melatonin can also bind some cytoplasmic proteins. Quinone reductase 2 is also referred to as cytoplasmic melatonin receptor MT3. The downstream effects of MT3 activation are unclear, but they are probably related to ROS scavenging by the enzyme [15], which converges with the antioxidative action of melatonin itself. Moreover, a direct interaction between melatonin and calmodulin can modify the activity of calmodulin-dependent kinase II (CaMKII) [16]. Collectively, these data suggest that the most characterized pathways of melatonin action include MT1 and MT2 membrane receptors and direct or indirect antioxidative effects.

Among other effects, melatonin has been found to provide a cardioprotective effect via either receptor-mediated signaling or its antioxidative properties [1,15,17,18]. Central and/or peripheral sympathetic nervous system inhibition is another effect of melatonin that may be involved in cardiac arrhythmia protection [19,20,21]. Moreover, melatonin reduced the incidence of malignant arrhythmias due to acute hypokalemia [17], which was attributed to the protection of electrical coupling protein connexin-43 (Cx43). Antiarrhythmic effects of melatonin in the condition of myocardial ischemia and reperfusion have been reported [22,23,24]. However, the molecular targets and/or mechanisms of the antiarrhythmic action of melatonin in the setting of ischemia or reperfusion are still not fully elucidated. Protection from reperfusion-induced malignant arrhythmias was mostly attributed to free radical scavenging and the anti-oxidative effects of melatonin [25]. These actions may preserve the sodium and potassium channel functions as well as calcium handling, thereby decreasing the susceptibility of the heart to arrhythmias [26]. Of interest, we have previously demonstrated that the antiarrhythmic and antioxidative effects of melatonin in acute reperfusion were not associated with each other [27]. These observations suggest that the antiarrhythmic effects of melatonin might be mediated by its receptor-dependent signaling pathways and have specific electrophysiological targets, such as ion channels.

The further studies performed in the chronic models showed that the antiarrhythmic melatonin effects were related to the improvement in myocardial impulse conduction mediated by the enhancement of sodium channels [28] and/or Cx43 channel function [29]. On the other hand, changes in ventricular repolarization were not associated with malignant arrhythmia incidence [27]. The sympatholytic effect of melatonin, though rarely observed, can also be discarded as the basis of antiarrhythmic action [30].

Acute effects of melatonin application were studied in anesthetized pigs [31,32] and ex vivo in perfused rat hearts [17]. These investigations demonstrated that melatonin reduced ventricular tachycardia and/or ventricular fibrillation (VT/VF) incidence. The experimental conditions (deep anesthesia and isolated heart) exclude the potential antiarrhythmic mechanism of melatonin via sympathetic inhibition. Instead, these studies, performed in different animal models, point out that melatonin mitigated conduction slowing in the affected myocardium via an as-yet-unknown mechanism.

Assuming similar acute and chronic melatonin action, one can suggest the sodium and Cx43 channels to be potential targets in both events. Indeed, acute administration of melatonin prevented hypokalemia-induced dephosphorylation of Cx43 and its redistribution from the intercalated disc to the lateral sides of cardiomyocytes that are pro-arrhythmic [17]. Consequently, melatonin significantly reduced hypokalemia-induced ventricular fibrillation. However, the sodium channel/current was not examined in this study. A transmembrane potential recording in isolated rat ventricular preparations demonstrated the melatonin-facilitated recovery of the resting membrane potential (RMP) during the reoxygenation period [27], which suggests the possible involvement of the inward rectifier current (IK1) in the melatonin action. Of note, the possible involvement of receptor-dependent signaling along with the antioxidative properties of melatonin has not been explored in this setting.

The aim of the current study was to explore the electrophysiological effects of the acute administration of melatonin in a rat myocardial reperfusion model. (I) We sought to reveal whether melatonin receptor-dependent or receptor-independent signaling is involved in the major electrophysiological effects of melatonin. (II) We aimed to identify the most relevant electrophysiological targets that may underlie the antiarrhythmic action of melatonin.

## 2. Results

### 2.1. In Vivo Heart Electrophysiology (Protocol A)

Following infusion of the tested agents (see protocol a in Figure 1), activation times (ATs) were shortened in the control (CONTROL) and melatonin (MEL) groups compared to the luzindole (LUZ) and luzindole + melatonin (MEL + LUZ) groups during reperfusion (Figure 2A). Dispersion of repolarization (DOR) in the melatonin groups (MEL and MEL + LUZ) was significantly decreased compared to the CONTROL and LUZ groups during reperfusion (Figure 2B).

### 2.2. Incidence of Ventricular Arrhythmias (Protocol A)

Reperfusion-induced VT/VF incidence was lower in the melatonin-treated animals as compared to the controls (MEL: 2 out of 12 animals vs. CONTROL: 13 out of 17 animals; *p* = 0.0015). The blockade of MT1/MT2 receptors with luzindole abolished the antiarrhythmic effect of melatonin (LUZ: 6 out of 10 animals; MEL + LUZ 9 out of 11 animals). In logistic regression analysis, ventricular tachycardia (VT), and VT/VF incidences were significantly associated with reperfusion AT (Table 1, Figure 2C).

### 2.3. Myocardial Conduction Velocity (Protocol B)

Figure 3 and Figure 4 display representative isochrone activation maps and mean longitudinal and transverse conduction velocities (CV_L_ and CV_T_, respectively) in different groups, respectively. Luzindole infusion caused a decrease in baseline CV_L_ as compared to the CONTROL group. Ischemia caused conduction slowing (both CV_L_ and CV_T_) in all groups, except MEL. In the CONTROL and LUZ groups, CV_L_ was decreased during reperfusion as compared to the baseline state. In both groups given melatonin (MEL and MEL + LUZ), CV_L_ at reperfusion did not differ from baseline.

### 2.4. Markers of Myocardial Oxidative Stress

Acute administration of melatonin did not affect the selected indices of oxidative stress in the ventricles of the rat heart subjected to ischemia and reperfusion (Figure 5). Accordingly, there was no difference in superoxide dismutase (SOD) activity as well as the content of 4-hydroxynonenal adducts (4HNE) or total glutathione (GSH) and total antioxidant capacity (TAC), between melatonin-treated and non-treated rats. Moreover, there were no statistically significant associations between the parameters of oxidative stress and the occurrence of reperfusion-induced malignant ventricular arrhythmias.

### 2.5. Cardiomyocyte Patch Clamp Measurements

In a normal Tyrode solution, the current amplitude of the sodium current (INa) was −82.12 ± 5.21 pA/pF at −30 mV. INa did not change during a 5 min exposure to 10 µM melatonin or 1 µM luzindole (Figure 6). In a normal Tyrode solution, the current amplitude of IK1 averaged 1.63 ± 0.12 pA/pF at −60 mV and −13.94 ± 1.11 pA/pF at −120 mV. During a 5 min exposure to melatonin (10 μM), the parameters of the IK1 current did not change. The MT1/MT2 receptor blocker luzindole at a concentration of 1 μM significantly reduced the amplitude of IK1 to 0.69 ± 0.16 pA/pF (*p* = 0.028) and −6.41 ± 0.92 pA/pF (*p* = 0.008) at −60 mV and −120 mV, respectively (Figure 6). The addition of melatonin (10 µM) to the solution containing luzindole (1 µM) partially reversed the effect of luzindole. Luzindole also depolarized the resting potential from −73 ± 2 to −63 ± 2 mV (*p* = 0.008) and increased the duration of the action potential at 100% repolarization from 136 ± 18 to 288 ± 32 ms (*p* = 0.008), while the luzindole/melatonin combination reversed the duration of the action potentials, but not the resting membrane potential.

### 2.6. Myocardial Cx43, P-Cx43^368^, and PKCƐ Protein Levels

The protein levels of phosphorylated Cx43 at serine 368 and PKCƐ, which directly phosphorylates Cx43, were significantly increased in the left ventricular myocardial tissue of the melatonin-treated animals, while the application of the MT1/MT2 receptor blocker luzindole significantly decreased myocardial levels of total Cx43, P-Cx43^368^, and PKCƐ. The combination of luzindole/melatonin partially normalized the blocking effect of luzindole (Figure 7).

## 3. Discussion

In the current study, we demonstrated the antiarrhythmic effect of melatonin, resulting in the reduced incidence of reperfusion VT/VF in an acute ischemic rat model. It is in line with the findings of the previous studies on acute melatonin action in various experimental models of ischemia and reperfusion [22,23,24,25,31,33]. Of interest, we ascertained that the short-lasting application of melatonin did not affect the selected markers of oxidative stress in the course of reperfusion, though the antioxidative properties of melatonin have been previously reported [34,35]. Likewise, the antioxidative effects of melatonin were not implicated in a porcine model of acute myocardial ischemia [31]. Moreover, in a chronic model of melatonin treatment [27], though the melatonin-related antioxidative effects were demonstrated, these effects were not associated with arrhythmia incidence. The absence of the significant antioxidant effect of melatonin was probably due to the fact that melatonin could not reach the affected myocardium during the ischemic episode, while the duration of reperfusion was not sufficient for melatonin to provide a significant action. Altogether, we can cautiously conclude that the obtained data do not support the potential role of the antioxidant properties in the realization of the antiarrhythmic action of melatonin in the setting of acute reperfusion.

In addition to the antioxidative effects, melatonin interacts with a number of various signaling pathways and activates Gi/o and Gq/11 protein-coupled receptors [36] as well as the phospholipase C (PLC) and protein kinase A (PKA) pathway [37,38,39]. However, their implication in the reperfusion setting in the context of cardiac arrhythmias has not been studied yet. The application of MT1/MT2 receptor blocker luzindole abolished the antiarrhythmic properties of melatonin. It suggests the implication of MT1/MT2-mediated signaling in the antiarrhythmic effect of melatonin. The downstream events following the receptor activation may involve several G-protein-related intracellular cascades [36], which were not explored in this study. The activation of PLC may be involved in the upregulation of PKCε that phosphorylates Cx43 at serine 368 [40], thereby conferring the protection of the Cx43 channels, mediating electrical conduction at the intercalated disc [41]. These alterations were abolished by luzindole and associated with the occurrence of reperfusion-induced malignant arrhythmias. Melatonin partly shared intracellular signaling pathways with adrenergic activation, and the sympatholytic effects of melatonin have been reported [20,21]. However, the effects of melatonin and luzindole on conduction demonstrated here are distinctive and are not expected to be yielded by the modification of adrenergic signaling. Collectively, the obtained data suggest the involvement of the MT1/MT2 receptor in the antiarrhythmic action of melatonin.

In the context of our findings, another issue that should be discussed is the impact of depolarization vs. repolarization on the antiarrhythmic effect of melatonin. Reentry mechanisms underlying the development of VT/VF in reperfusion settings can be based on the changes in both depolarization (CV) and repolarization (DOR) characteristics [42], and melatonin modified both of them. It prevented the ischemic decrease in CV and reduced DOR. However, luzindole did not affect DOR. This observation implies the involvement of the MT1/MT2-independent mechanism underlying the melatonin effects on repolarization. On the other hand, luzindole increased AT delay, slowed CV, and abolished the antiarrhythmic effect of melatonin. Furthermore, the VT/VF incidence was associated with the AT delay, but not DOR, in a logistic regression analysis. A similar effect of melatonin has been demonstrated in a chronic model of melatonin treatment, where melatonin modified both depolarization and repolarization processes, but only the depolarization changes were associated with the antiarrhythmic action of melatonin, while the effect on repolarization (but not on depolarization) correlated with antioxidant activity [27]. Also, the data obtained in the present study correspond to the findings in the porcine model of acute ischemia with the application of melatonin [31,32]. Collectively, the present findings suggest that the antiarrhythmic effects of melatonin were related to the changes in the depolarization process.

For the elucidation of the acute antiarrhythmic effect of melatonin, its electrophysiological targets should be identified. Our study in the chronic model of melatonin treatment [28] demonstrated that conduction enhancement was caused by the melatonin-related increase in the sodium channel protein expression (Scn5a/Nav1.5) resulting in the augmentation of the sodium current INa. We tested INa in the present study, having in mind the hypothesis that similar effects might be induced by acute melatonin application. However, neither melatonin nor luzindole affected INa, which allows us to reject this current as the target of acute melatonin action.

Other previous studies from our group [17,29,43] demonstrated that melatonin can influence Cx43 protein levels, topology, and phosphorylation, preventing adverse remodeling in different pathological states. In the present study, we also demonstrated that melatonin and luzindole caused the opposite changes in the content of the P-Cx43^368^ variant and amount of PKCε, the enzyme related to the P-Cx43^368^ variant formation. In chronic conditions, the increase in the P-Cx43^368^ variant has been associated with the protection against malignant arrhythmias, stabilization of gap junctions in the intercalated disks, and prevention of connexin lateralization [41]. Collectively, the effects of melatonin and luzindole on Cx43 properties can underlie the observed changes in conduction and VT/VF incidence.

Melatonin application facilitated the recovery of the RMP in rat right ventricular preparations during reoxygenation after hypoxia [27]. This finding implies that the IK1 current maintaining the resting membrane potential might be involved in the realization of the melatonin effect, especially at reperfusion. We did not observe changes in IK1 under melatonin application, but the MT1/MT2 receptor blockade with luzindole significantly reduced it, which suggests the probable involvement of receptor-dependent signaling in the observed melatonin and luzindole effects. G-protein coupled receptors (including melatonin receptors) can regulate a subfamily of inward-rectifier potassium channels [44,45], which includes the IK1 current, and the observed effect of luzindole corresponds to this mechanism.

Luzindole, a golden-standard blocker for MT1 and MT2 receptors, has been reported to affect cardiac connexins [17] and other potassium channels in the nervous system [46]. Luzindole was shown to demonstrate an agonistic activity for the cAMP-dependent branch of the melatoninergic pathway [47]. An increased cAMP level and activation of PKA influenced the Kir channels and decreases their activities in the cardiomyocytes [48]. Also, luzindole and 4P-PDOT competitively block MT1 melatonin receptors, and both are inverse agonists in systems with constitutively active MT1 receptors [10,49,50,51,52]. These luzindole effects could lead to a reduction in the amplitude of the IK1 current shown in this study.

The changes in the IK1 current and the resting membrane potential can indirectly modify the sodium current. The preservation of the RMP on a relatively hyperpolarized level maintains sodium channel availability, which in turn favors conduction, and vice versa. This gives a plausible explanation for the observed effects of melatonin or luzindole on CV. Since the sodium channels are expressed differently in the lateral and polar regions of the cardiomyocytes, and the greatest sodium current has been observed at the intercalated discs [53], the longitudinal conduction, which is largely related to the intercalated discs, is expected to experience more influence from the changes in IK1. It probably accounts for the changes in the longitudinal but not the transverse CV with the application of melatonin or luzindole.

Limitations. (I) There are some limitations concerned with the nature of this experimental study. The rat electrophysiological phenotype is quite different from that in humans and the used experimental model of short ischemia followed by rapid reperfusion is far from clinical conditions. Moreover, the sizes of the animal groups were small. Thus, the results of the present study should be interpreted and extrapolated to humans cautiously. (II) The size of the ischemic zone was not assessed in this study since the excised hearts after in vivo experiments were used in further analyses (patch-clamp electrophysiology, Western blot, and oxidative stress evaluation), and therefore we could not evaluate its contribution to arrhythmogenesis. However, we used the established technique for ligating the anterior descending branch of the left coronary artery in the region of its proximal third for 5 min. This method has been proven to yield standard results, as the previous studies using the same experimental model [27,30,54,55] revealed no differences in the ischemic area size between animals and no associations between the ischemic area size and the arrhythmic outcomes. (III) The observed effect of the melatonin receptor blocker luzindole alone might be caused by the pre-existing activity of the melatoninergic pathways in the myocardial tissues. However, this kind of activity could not be characterized in our study, and some mechanistical issues concerning the melatonin/luzindole action remain unresolved.

## 4. Materials and Methods

### 4.1. Animals and Experimental Design

In the present study, we used two in vivo protocols (Figure 1). In the first protocol, we tested the antiarrhythmic potential of melatonin in a setting mimicking a clinical situation when the treatment is performed in ischemia when life-threatening arrhythmias are expected. This protocol was also used to differentiate between receptor-dependent and receptor-independent (antioxidative) mechanisms of the potential antiarrhythmic effects of melatonin. Therefore, tested substances (melatonin, MT1/MT2 receptor luzindole, or their combination) were administered immediately before reperfusion, and after completion of experiments, the hearts were taken for the assessment of oxidative stress parameters and connexin measurements. In the second protocol, the same tested substances were introduced before coronary occlusion to allow their delivery to the entire myocardium. This protocol permitted evaluation of melatonin and luzindole effects in normal and ischemic tissues. After these experiments, the hearts were taken for myocyte isolation and patch-clamp measurements (the placebo groups only). Data obtained from several animals of different groups were discarded due to their technical unsuitability for analysis (poor signal-to-noise ratio, no-reperfusion phenomenon, distorted electrogram signals).

Experiments were performed in a total of 86 Wistar male rats (3-month-old, 250–300 g). The animals were provided by the Collection of Experimental Animals of the Institute of Biology, Federal Research Center, Komi Scientific Center, Ural Branch of the Russian Academy of Sciences. The study was approved by the ethical committee of the Institute of Physiology of the Komi Science Centre, Ural Branch of Russian Academy of Sciences (approval 19 February 2018, amendment 19 September 2018), and all experiments conformed to the Guide for the Care and Use of Laboratory Animals, 8th Edition published by the National Academies Press (US), 2011, and the guidelines from Directive 2010/63/EU of the European Parliament.

The rats were anesthetized with intramuscular injection of zoletil (Virbac S.A., Carros, France, 15 mg/kg) and xylazine (Interchemie, Castenray, The Netherlands, 0.1 mg/kg), and mechanical ventilation was established (for in vivo experiments). The heart was exposed by a mid-sternal incision and kept warm (37–38 °C). Myocardial ischemia was induced by the left anterior descending coronary artery (LAD) ligation (coated braided polyester, 5–0, Ti-Cron, Cardiopoint, Covidien, Santo Domingo, Dominican Republic) in the region of its proximal third for 5 min followed by reperfusion. This technique was used in a number of our previous experimental studies as an established arrhythmia model, which provides a sufficient number of endpoints (reperfusion ventricular tachycardia (VT) and ventricular fibrillation (VF) episodes) [27,30,54,55]. Cautiously speaking, short episodes of vasospastic angina and spontaneous recanalization after thrombus formation could be considered clinical counterparts of the present model, although differences between the experimental and clinical conditions should be taken into account.

Placebo (saline), melatonin (4 mg/kg), luzindole (0.4 mg/kg), or melatonin + luzindole combination were intravenously infused (0.25–0.30 mL) before ischemia (to evaluate changes in the normal, ischemic, and reperfused myocardium) or before reperfusion (to evaluate reperfusion arrhythmia incidence and its association with other parameters). The dosage of the melatonin treatment in vivo was the same as in our previous work [31] for the sake of consistency. For the in vitro study, the concentration of melatonin was selected in accordance with the published data [23,56]. Luzindole, a blocker of MT1/MT2 receptors was applied to recognize receptor-dependent and receptor-independent signaling pathways. Arrhythmia incidence, average epicardial activation time, and ventricular conduction velocity (CV) were measured in vivo using epicardial mapping and ECG recording. Under deep anesthesia at the end of each in vivo experiment, the animals were euthanized by rapid heart excision. Cardiac tissue was sampled for further analysis of connexin-43 (Cx43), and its variant P-Cx43^368^, which stabilizes Cx43 channel conduction at the intercalated disc, and PKCε is known to phosphorylate Cx43 at serine 368. To identify the electrophysiological targets of melatonin, we determined the parameters of the transmembrane potentials, sodium current (INa), and inward rectifier potassium current (IK1) in patch-clamp studies in isolated cardiomyocytes.

### 4.2. Epicardial Mapping

The procedure of epicardial mapping has been described elsewhere [28,55]. In brief, in 12 animals of the melatonin group (MEL), 10 animals of the luzindole group (LUZ), 11 animals of melatonin + luzindole (MEL + LUZ) group, and 17 animals of the control group (CONTROL), unipolar electrograms were recorded in the left ventricle (LV) using a 64-lead array. In each lead, activation time (AT) and repolarization time (RT) were determined as the instants of dV/dt min during QRS complex and dV/dt max during T-wave, respectively. Activation–repolarization interval (ARI), a surrogate for action potential duration, was calculated as ARI = RT − AT. Dispersion of repolarization (DOR), which is generally known to be a prerequisite of the unidirectional conduction block and therefore of reentrant ventricular tachyarrhythmias, was calculated as the difference between the maximal and minimal RTs throughout all leads in the same cardiac cycle in the same recording. VT/VFs were assessed at reperfusion after 5 min ischemia induced by LAD ligation.

In a separate set of 36 rats exposed to MEL (n = 9), LUZ (n = 10), and MEL + LUZ (n = 9) as well as CONTROLS (n = 8), CV was measured using isochrone activation mapping under electrical stimulation (400 bpm, 2 mA, 2 ms) in the middle of the LV. Conduction velocity was calculated as the distance traversed across the multielectrode array, divided by the time difference between the ATs at the pacing site and the earliest activation time at the opposite side of the multielectrode array, i.e., CV = traversed distance/(AT opposite − AT pacing site). Longitudinal (CV_L_) and transversal (CV_T_) velocities were determined in mutually perpendicular directions as the maximal and minimal conduction velocities, respectively.

### 4.3. Whole-Cell Patch-Clamp Recording

The previously described isolation and recording procedures were used [28]. Cardiomyocytes were enzymatically isolated from the excised hearts of 10 control animals after in vivo electrophysiological examination. Whole-cell patch-clamp experiments were performed with the Axopatch 200B (Axon instrument, Burlingham, CA, USA) at room temperature (20–24 °C). INa current was recorded with external Cs-based low-Na+ solution in the presence of 2 × 10 − 5 M nifedipine in bath solution to block ICaL, as described earlier [28]. Resistance of the pipettes was 1.6 ± 0.3 MΩ.

To record the IK1 current in the voltage clamp mode and action potential in the current-clamp mode, the pipettes were filled with a solution containing (mM) 140 KCl, 25 HEPES, 3 MgATP, 0.4 NaGTP, 0.5 EGTA; pH 7.2. A bath chamber was perfused whit solution containing (mM): 150 NaCl, 5.4 KCl, 1.8 CaCl_2_, 1.2 MgCl_2_, 10 glucose, 10 HEPES, pH 7.4. The IK1 current was evoked by a change in membrane potential according to a linear protocol from +60 to −120 mV. IK1 was recorded in the presence of 10 μM nifedipine and 2 mM 4-aminopyridine in solution. To determine the magnitude of the leakage current, 3 mM BaCl2 was added to the extracellular solution.

To evaluate the acute effect of melatonin on ionic currents, 10 μM melatonin and 1 µM luzindole were added to each of the extracellular solutions.

### 4.4. Determination of Oxidative Stress Markers

Parameters of oxidative stress were evaluated by the blinded way manner in the hearts of the control and melatonin-treated animals (n = 10 in each group). The procedures were previously described in detail [27]. In brief, total glutathione (GSH), superoxide dismutase activity (SOD), total antioxidative capacity (TAC), and 4-hydroxyneonal adducts (HNE) were detected by the Superoxide Dismutase Activity Assay, HNE Adduct Competitive ELISA Kit, and Total Antioxidant Capacity Assay Kit (Cell Biolabs, San Diego, CA, USA), according to the manufacturers’ protocols. The 1× phosphate-buffered saline (PBS) (pH 7.4) was utilized in all experiments. The samples were clarified by centrifugation (MPW-260R, MPW, Warsaw, Poland) and were measured by ELISA reader LT-5000MS (Labtech International Ltd., Uckfield, UK). Experiments were performed in duplicates, and their average values were used for statistical analysis.

### 4.5. Cx43 and PKC ε Levels Assessed by Western Blot Analysis

As was described previously [57,58], left ventricular tissue was homogenized in SB20 lysis buffer (20% SDS, 10 mmol/L EDTA, and 100 mmol/L Tris, pH 6.8) and diluted in the Laemmli sample buffer. Then, 20 μg of proteins per lane were separated in 10% SDS-PAGE acrylamide gels, transferred onto a nitrocellulose membrane (0.2 μm pore size, Advantec, Tokyo, Japan), and blocked with 5% fat-free milk. Subsequently, the membranes were incubated in primary antibodies (1:Cx43; 1:5000; C6219; Sigma-Aldrich, St. Louis, MI, USA; 2:pCx43^368^; 1:1000; sc-101660; Santa Cruz Biotechnology, Dallas, Texas, USA; 3:PKCε; 1:1000; sc-214; Santa Cruz Biotechnology, Dallas, TX, USA) followed by incubation in secondary antibodies (1:2000; 7076C; Cell Signaling Technology, Denver, CO, USA). For protein visualization, an enhanced luminol-based chemiluminescent was used. Observed proteins were normalized to GAPDH and densitometrically analyzed using Carestream Molecular Imaging Software (version 5.0, Carestream Health, New Haven, CT, USA).

### 4.6. Statistical Analysis

Statistical analyses were performed using GraphPad Prism version 8.0. The Shapiro–Wilk test for data distribution was used. Differences between groups were evaluated using one-way ANOVA with Tukey or Dunnet post hoc tests, when appropriate. Logistic regression analysis was used to test associations between arrhythmic outcomes and electrophysiological parameters. Chi-square test was used to compare arrhythmia incidence between different groups. All data are presented as mean ± SEM, and the differences were considered significant at *p* < 0.05.

## 5. Conclusions

The present investigation confirmed the previously observed antiarrhythmic effects of melatonin with its acute application during myocardial reperfusion. We further elucidated, at least partially, the mechanisms of this action, which appeared to be independent of the known antioxidative melatonin properties but were associated with the MT1/MT2 receptor-dependent signaling pathway. The regulation of intercellular coupling via connexin43 and resting membrane potential by the IK1 current were demonstrated to be the most probable electrophysiological targets of melatonin action. The obtained data suggest that melatonin signaling is distinctly related to myocardial conduction properties. These observations might promote the search for novel treatment modalities, especially those targeted to the applications in ischemic conditions, where many other approaches appear ineffective.

## Figures and Tables

**Figure 1 ijms-24-11931-f001:**
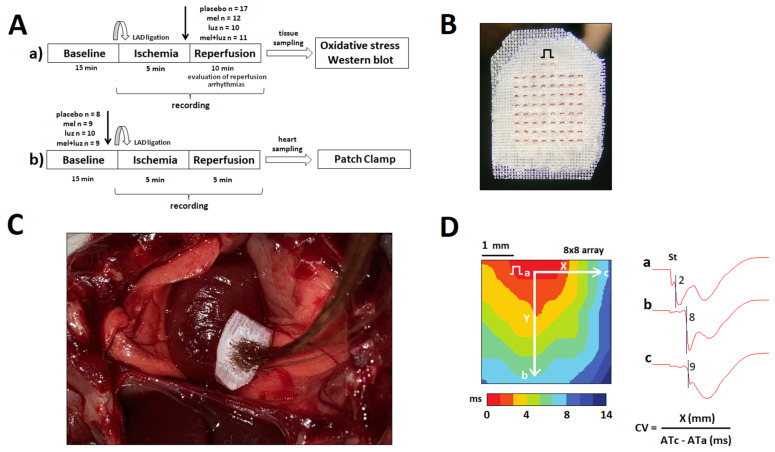
Methods: Panel (**A**): Schematic presentation of the study design. The tested substances were administered just before reperfusion (protocol a) and in baseline (protocol b). After in vivo electrophysiological study, the heart was rapidly excised to be used for the measurements of oxidative stress parameters, Western blot analysis of Cx43, its variant P-Cx43^368^, and protein kinase C Epsilon (PKCƐ), as well as for in vitro electrophysiological study (placebo group only) in protocols a and b, respectively. Panel (**B**): The lead terminals’ surface of the recording plate (4 × 4 mm, 64 leads, 2 stimulating electrodes). Electrical step symbol indicates a pacing site. Panel (**C**): View of the in vivo heart preparation with the recording plate positioned on the left ventricle. Panel (**D**): Representative examples of the baseline activation map and the electrograms recorded at sites a, b, and c, which were used for calculation of the longitudinal and transverse conduction velocity. “St” indicates a pacing artifact.

**Figure 2 ijms-24-11931-f002:**
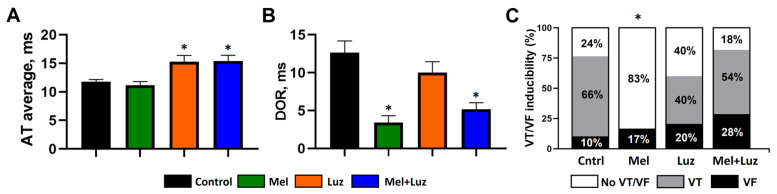
Electrophysiological parameters in the left ventricle at reperfusion with the infusion of placebo (Control), melatonin (Mel), luzindole (Luz), and melatonin + luzindole (Mel + Luz) prior to reperfusion (see protocol a). Bar graphs depict average AT at reperfusion (panel (**A**)) and DOR at reperfusion (panel (**B**)). Data are presented as mean ± SEM. Panel (**C**) depicts the incidence of ventricular tachycardia (VT) and ventricular fibrillation (VF). * *p* < 0.05 vs. control.

**Figure 3 ijms-24-11931-f003:**
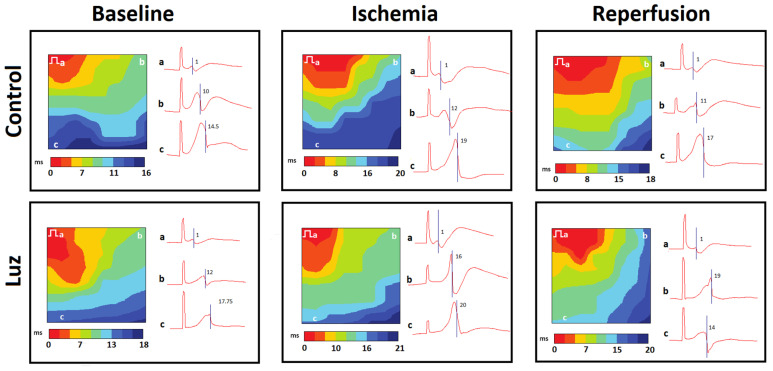
Representative left ventricular isochrone activation maps and unipolar electrograms at the specified sites (a, b, c) during electrical pacing of the control- and luzindole-treated rats. Electrical step symbol indicates the site of pacing. Note the longer periods (maximal values on scales) needed for activation spread across the mapped area with luzindole application.

**Figure 4 ijms-24-11931-f004:**
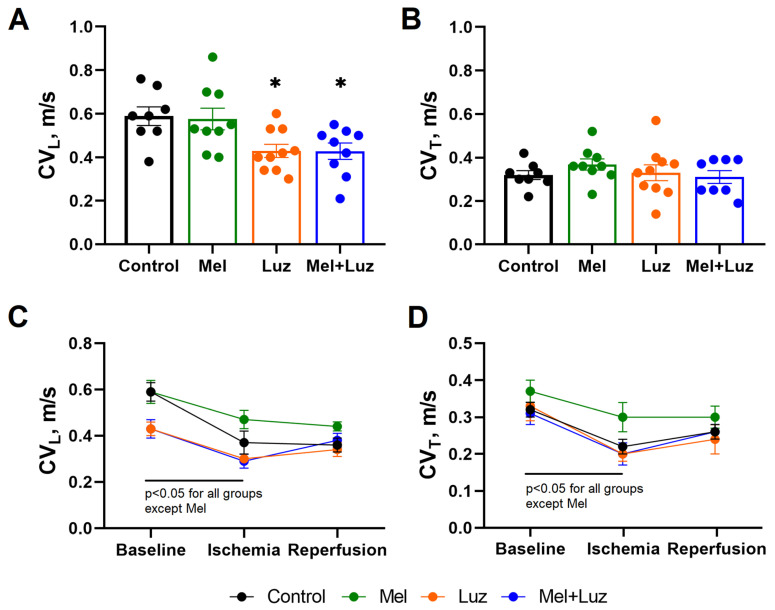
Conduction velocity (CV) in the left ventricle in the control, melatonin, luzindole, and melatonin + luzindole groups (mean ± SEM). Melatonin prevents ischemia CV slowing. All tested substances were infused at baseline (see protocol b in Figure 1). Bar graphs depict the longitudinal (CV_L_, panel (**A**)) and transverse (CV_T_, panel (**B**)) at the baseline state. Linear graphs depict the changes in the CV_L_ (panel (**C**)) and the CV_T_ (panel (**D**)) at ischemia and reperfusion. * *p* < 0.05 vs. control.

**Figure 5 ijms-24-11931-f005:**
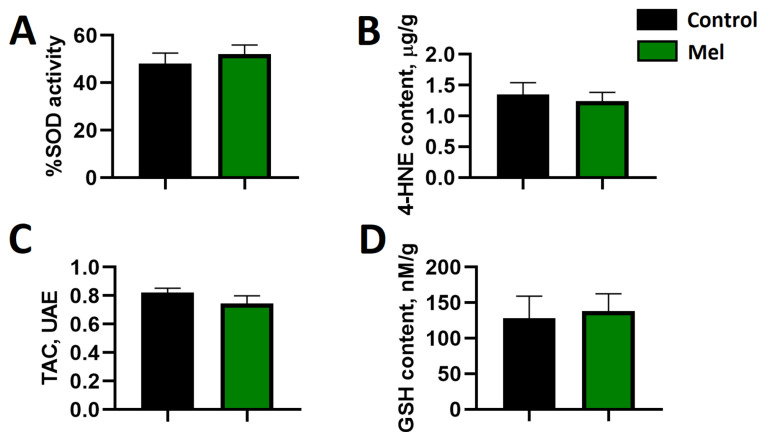
Effect of melatonin on the levels of superoxide dismutase activity (SOD, panel (**A**)), 4-hydroxynonenal adducts content (4-HNE, panel (**B**)), total antioxidant capacity (TAC, panel (**C**)), and total glutathione content (GSH, panel (**D**)). No significant differences were found.

**Figure 6 ijms-24-11931-f006:**
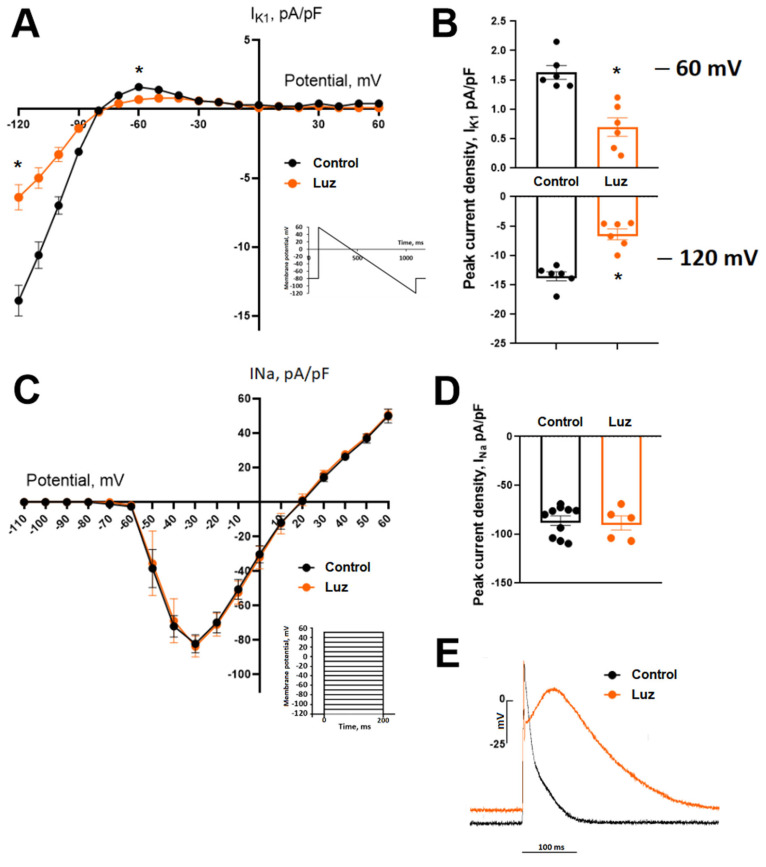
Patch-clamp studies of the IK1 (panels (**A**,**B**)) and INa (panels (**C**,**D**)) ionic currents in the control cardiomyocytes and with the application of luzindole. Voltage–current curves (panels (**A**,**C**)) and peak current densities (panels (**B**,**D**)) for both currents are presented. Luzindole caused a decrease in the IK1 current and no effect on INa. Comparison of peak current densities of IK1 in the control and luzindole groups for outward potassium current (upper) and inward potassium current (lower) (panel (**B**)). Comparison of voltage–current curves (panel (**C**)) and peak current densities (panel (**D**)) of INa in the control and luzindole groups; no significant differences were observed. Effect of luzindole on action potential in the current-clamp mode (representative recordings, panel (**E**)). * *p* < 0.0005 vs. control.

**Figure 7 ijms-24-11931-f007:**
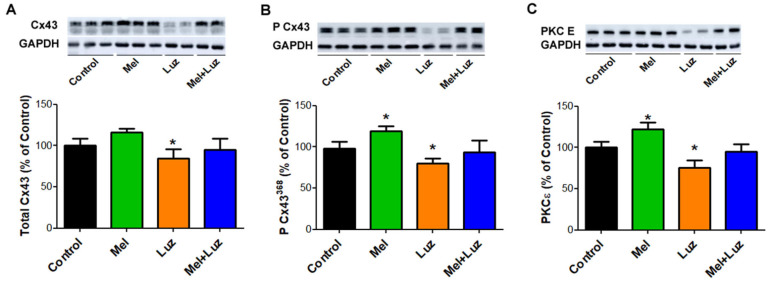
Immunoblot demonstration and protein levels of myocardial Cx43 (**A**), functional P-Cx43^368^ variant (**B**), and PKCε (**C**) in experimental rats. Note the tendency of melatonin to increase these protein levels, while luzindole abolished this effect. Cx43—connexin 43, P-Cx43^368^—Connexin 43 variant phosphorylated at serine 368, PKCε- protein kinase C Epsilon, * *p* < 0.05 vs. Control. Data are presented as means ± SD, n = 6 per group.

**Table 1 ijms-24-11931-t001:** Associations between arrhythmic outcomes and electrophysiological mapping characteristics.

Outcome	AT		DOR	
	Odd Ratio	*p*	Odd Ratio	*p*
VT	1.591 95% CI 1.061–2.386	0.025	0.932 95% CI 0.793–1.096	0.397
VF	1.052 95% CI 0.731–1.516	0.784	1.100 95% CI 0.883–1.370	0.395
VT/VF	1.695 95% CI 1.046–2.748	0.032	1.088 95% CI 0.922–1.284	0.316

## Data Availability

The data presented in this study are available on request from the corresponding author. The data are not publicly available due to policy of the Institute of Physiology of the Komi Science Centre, Ural Branch of Russian Academy of Sciences.

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
