# Peer review of "Blockade of Melatonin Receptors Abolishes Its Antiarrhythmic Effect and Slows Ventricular Conduction in Rat Hearts"

_ijms, 2023, doi:10.3390/ijms241511931_

Round 1

Reviewer 1 Report

The manuscript presents a study where the authors assessed the effects of melatonin and a MT-receptor blocker luzindole in an experimental acute ischemia/reperfusion model. Multiple physiologic effects were analyzed and summarized.

The text will need major revision. Some sections still contain default entries from the manuscript template. The text is also out of order, the methods section should be presented prior to results and discussion. Each abbreviation should be explained at their first appearance (such as ROS, GPCR in the introduction section).

Why are the sample sizes uneven for placebo, mel, luz and mel+luz groups? Were unsuccessful measurements discarded? For better visual comparison, consider reformatting Figure 2C to % instead of absolute case numbers.

Based on the findings, luzindole has an intrinsic effect (Ik1), which may complicate its assessment as a melatonin antagonist. The effect is more than just simple MT1/2 blockade. Fig2A and 2B suggest either a separate effect or incomplete blockade of melatonin.

Are the doses used in vivo and the concentration used in vitro represent physiologic ranges of melatonin?

Longitudinal, but not transverse conduction velocity effects were noted with the studies agents. Fig 4A, B. Discuss the potential mechanisms.

Room temperature was used for the patch clamp measurements. This may have affected ion current characteristics or melatonin/luziondole effects. If these effects are insensitive to temperature changes, cite the relevant studies.

No significant antioxidant effect of melatonin was observed in this acute study. How does it correlate with data from the cited studies where some antioxidant effect was observed?

The manuscript will require extensive editing and several questions will need to be answered to clarify the results of the study.

Author Response

REVIEWER: 1

Comment 1:

Some sections still contain default entries from the manuscript template. The text is also out of order, the methods section should be presented prior to results and discussion. Each abbreviation should be explained at their first appearance (such as ROS, GPCR in the introduction section).

Response:

Thank you! In the revised version, we removed unnecessary template entries and expanded all abbreviation at first mention. However, the position of the Methods section after Results and Discussion is according to the IJMS template, therefore we left it as is.

Comment 2:

Why are the sample sizes uneven for placebo, mel, luz and mel+luz groups? Were unsuccessful measurements discarded?

Response:

This inequality was due to that the data from several animals could not be technically analyzed. The causes of these problems included the poor signal/noise ratio, no-reperfusion phenomenon, and distorted or low-amplitude QRST-complexes in ventricular electrograms. Also, the heart preparations from the larger placebo group were used for in vitro experiments. An appropriate information was included in the Methods section.

Lines 355-358:

After these experiments, the hearts were taken for myocyte isolation and patch-clamp measurements (the placebo groups only). Data obtained from several animals of different groups were discarded due to their technical unsuitability for analysis (poor signal-to-noise ratio, no-reperfusion phenomenon, distorted electrogram signals).  

Comment 3:

For better visual comparison, consider reformatting Figure 2C to % instead of absolute case numbers.

Response:

Amended as suggested.

Figure 2. Electrophysiological parameters in the left ventricle at reperfusion with the infusion of placebo (Control), melatonin (Mel), luzindole (Luz) and melatonin+luzindole (Mel+Luz) prior to reperfusion (see protocol a). Bar graphs depict average AT at reperfusion (panel A) and DOR at reperfusion (panel B). Data are presented as mean ± SEM. Panel C depicts the incidence of ventricular tachycardia (VT) and ventricular fibrillation (VF). * p<0.05 vs. control.

Comment 4:

Based on the findings, luzindole has an intrinsic effect (Ik1), which may complicate its assessment as a melatonin antagonist. The effect is more than just simple MT1/2 blockade.

Response:

We appreciate the comment of the reviewer. Indeed, the effect of luzindole deserves consideration. Unfortunately, we were limited in our capabilities to directly evaluate it.

In the revised version, we included an additional paragraph to the Discussion section:

Lines 305-313:

Luzindole, a golden-standard blocker for MT1 and MT2 receptors, has been reported to affect cardiac connexins [17] and other potassium channels in the nervous system [46]. Luzindole was shown to demonstrate an agonistic activity for the cAMP-dependent branch of the melatoninergic pathway [47]. An increased cAMP level and activation of PKA influenced the Kir channels and decreases their activities in the cardiomyocytes [48]. Also, luzindole and 4P-PDOT competitively block MT1 melatonin receptors, and both are inverse agonists in systems with constitutively active MT1 receptors [10, 49-52]. These luzindole effects could lead to a reduction in the amplitude of the IK1 current shown in this study.

Also, we extended the limitation part:

Lines 337-341:

(III) The observed effect of the melatonin receptor blocker luzindole alone might be caused by the pre-existing activity of the melatoninergic pathways in the myocardial tissues. However, this kind of activity could not be characterized in our study, and some mechanistical issues concerning the melatonin/luzindole action remain unresolved.

Comment 5:

Fig2A and 2B suggest either a separate effect or incomplete blockade of melatonin.

Response:

Indeed, we consider that the melatonin effect on dispersion of repolarization was receptor-independent. Previously [Sedova et al., Int J Mol Sci, 2019], we showed that chronic melatonin treatment modified both depolarization and repolarization characteristics, and the antiarrhythmic action of melatonin was associated only with depolarization changes. In that study, the changes of repolarization (but not of depolarization) were associated with the changes in superoxide dismutase activity. Also, our preliminary patch-clamp results (not included in this manuscript) indicate that melatonin affects the calcium current, and this effect is independent of luzindole. Collectively, these considerations suggest that the melatonin effects on the repolarization phase of the action potential could be realized via receptor-independent mechanism.

In the present version, we extended the discussion of this point.

Lines 268-274:

A similar effect of melatonin has been demonstrated in a chronic model of melatonin treatment, where melatonin modified both depolarization and repolarization processes, but only the depolarization changes were associated with the antiarrhythmic action of melatonin, while the effect on repolarization (but not on depolarization) correlated with antioxidant activity [27]. Also, the data obtained in the present study correspond to the findings in the porcine model of acute ischemia with the application of melatonin [31, 32].

Comment 6:

Are the doses used in vivo and the concentration used in vitro represent physiologic ranges of melatonin?

Response:

The short answer is “No”, it is higher.

The dose of melatonin was chosen to be in accordance with previous investigations. In the present study, the doses of melatonin used in vivo and in vitro exceed the physiologic ranges of melatonin, but are not toxic [Sugden, J Pharmacol Exp Ther, 1983; Mantle el al., Sleep Med X, 2020]. It has been reported that significant changes of physiological parameters by melatonin in vivo is found at a dose 5-10 mg/kg i.v. [Lee et al., J Pineal Res, 2002].  Our previous study with intravenous melatonin infusion in pigs [Tsvetkova et al., Int J Mol Sci, 2021] demonstrated electrophysiological and antiarrhythmic effects at a dose of 4 mg/kg.

The concentration of melatonin to in vitro study was also selected according to the published data. Perfusion solution containing 10 uM of melatonin had an effect on cardiomyocytes and reduced the incidence of ventricular fibrillation in vitro [Tan et al., J pineal Res, 1998], also melatonin improved cardiomyocyte survival in cardiotoxicity models [Govender et al., Toxicol Appl Pharmacol, 2018].

This information is now provided in the Methods section.

Lines 384-387:

The dosage of the melatonin treatment in vivo was the same as in our previous work [31] for the sake of consistency. For the in vitro study, the concentration of melatonin was selected in accordance with the published data [23, 56].

Comment 7:

Longitudinal, but not transverse conduction velocity effects were noted with the studies agents. Fig 4A, B. Discuss the potential mechanisms.

Response:

In the present version, we extended the Discussion section:

Lines 314-323:

The changes in IK1 current and the resting membrane potential can indirectly modify the sodium current. The preservation of the RMP on a relatively hyperpolarized level maintains sodium channel availability, which in turn favors conduction, and vice versa. This gives a plausible explanation for the observed effects of melatonin or luzindole on CV. Since the sodium channels are expressed differently in lateral and polar regions of the cardiomyocytes, and the greatest sodium current has been observed at the intercalated discs [53], the longitudinal conduction, which is largely related to the intercalated discs is expected to experience more influence from the changes in IK1. It probably accounts for the changes in the longitudinal but not transverse CV with the application of melatonin or luzindole.

 Comment 8:

Room temperature was used for the patch clamp measurements. This may have affected ion current characteristics or melatonin/luziondole effects. If these effects are insensitive to temperature changes, cite the relevant studies.

Response:

Indeed, ionic currents are sensitive to temperature, but the measurements under room temperature are quite common and the data obtained here can be compared with others. Another reason to do it is that sodium current is very high and could be hardly evaluated in realistic conditions. Lowering the ambient temperature is one of the methods to decrease it to ensure more reliable measurements.

Comment 9:

No significant antioxidant effect of melatonin was observed in this acute study. How does it correlate with data from the cited studies where some antioxidant effect was observed?

Response:

We believe that this was due to that melatonin was added after ischemia had been induced and therefore it could not reach the affected area. It is noteworthy that the reperfusion period in this study was rather short, which probably was not sufficient for melatonin to provide the antioxidative effects. Usually, such effects are observed when melatonin is administered chronically, or immediately before ischemia, or with a pretty long period of reperfusion. For our point, it was important to demonstrate electrophysiological and antiarrhythmic effects of melatonin without antioxidative action.

The discussion of this point is extended.

Lines 234-237:

The absence of the significant antioxidant effect of melatonin was probably due to the fact that melatonin could not reach the affected myocardium during the ischemic episode, while the duration of reperfusion was not sufficient for melatonin to provide a significant action.

Reviewer 2 Report

Dear Sirs,

Thank you for sending the manuscript entitled: „Blockade of Melatonin Receptors Abolishes Its Antiarrhythmic Effect and Slows Ventricular Conduction in Rat Hearts”.

The study presented here concerns an in vivo analysis on an animal model of the effect of melatonin on the risk of dangerous ventricular arrhythmias. Due to the nature of the study, a small group of probands was included in the analysis. This is a limitation of the study that may undermine its promising conclusions. Other than that, in my opinion, the manuscript is methodologically correctly written and the conclusions remain consistent with the evidence and arguments presented.

It seems that lines: 35-42 and 121-130 - a piece of guidance for authors has been included.

It also seems to me that the layout of the manuscript should be improved; it is currently as follows: 1) Introduction 2) Results 3) Discussion 4) Materials and methods 5) Conclusions 6) Limitations

Yours sincerely

Reviewer

Author Response

REVIEWER: 2

Comment 1:

Due to the nature of the study, a small group of probands was included in the analysis. This is a limitation of the study that may undermine its promising conclusions.

Response:

We included this point in the limitation part:

(I) There are some limitations concerned with the nature of this experimental study. The rat electrophysiological phenotype is quite different from that in humans and the used experimental model of short ischemia followed by rapid reperfusion is far from clinical conditions. Moreover, the sizes of the animal groups were small. Thus, the results of the present study should be interpreted and extrapolated to humans cautiously.

Comment 2:

It seems that lines: 35-42 and 121-130 - a piece of guidance for authors has been included.

Response:

Yes, of course. Corrected.

Comment 3:

It also seems to me that the layout of the manuscript should be improved; it is currently as follows: 1) Introduction 2) Results 3) Discussion 4) Materials and methods 5) Conclusions 6) Limitations

Response:

The position of the Methods after the Discussion is according to the IJMS template. The limitation section is now moved to the Discussion section. Thank you for the notice!

Reviewer 3 Report

Aleksandra V. Durkina, et al. presented an article about melatonin and its anti-arrhtyhmia effect. The followings are some comments to be addressed.

Major comments

1.         In the Introduction section, there was a part not related to this article (Page 3, line 121 to 130). Please delete it.

2.         The author used luzindole to block MT1/MT2 receptor. According to most of the study results, luzindole itself would produce effect on electrophysiological characters. The findings suggested that there was pre-existed melatonin in the examined tissue. Otherwise, blockage of melatonin should not produce such effect. What is the concentration or what is the effect of the pre-existed melatonin should be further clearly expressed.

3.         The study mentioned DOR for multiple times. However, the definition of DOR was not disclosed. The impact of DOR should also be more offered.

4.         The format of this article was to place the Method section between Discussion and Conclusion. However, there were several abbreviated terms was used in the Results section, while the definition was in the Method section. Please define the abbreviated terms when they were firstly appeared in the whole article.

Author Response

REVIEWER: 3

Comment 1:

In the Introduction section, there was a part not related to this article (Page 3, line 121 to 130). Please delete it.

Response:

Yes, thank you very much! Corrected.

Comment 2:

The author used luzindole to block MT1/MT2 receptor. According to most of the study results, luzindole itself would produce effect on electrophysiological characters. The findings suggested that there was pre-existed melatonin in the examined tissue. Otherwise, blockage of melatonin should not produce such effect. What is the concentration or what is the effect of the pre-existed melatonin should be further clearly expressed.

Response:

We appreciate the comment of the reviewer. Indeed, the effect of luzindole alone deserves consideration. Unfortunately, we were limited in our capabilities to directly evaluate it. Specifically, we cannot confirm persistence of melatonin in the tissues in our experimental models. However, it is known that luzindole and 4P-PDOT competitively block MT1 melatonin receptors (in concentrations higher than 300 nM), and both are inverse agonists in systems with constitutively active MT1 receptors (Cecon et al., 2018; Update on Melatonin Receptors: IUPHAR Review 20, 2016; Johansson et al., 2019; Slominski et al., 2012; Stauch et al., 2019). This could partially explain the results found when luzindole was administered alone, but a complete prove is beyond the reach of the current study.

We included an additional paragraph in the Discussion section:

Lines 305-313:

Luzindole, a golden-standard blocker for MT1 and MT2 receptors, has been reported to affect cardiac connexins [17] and other potassium channels in the nervous system [46]. Luzindole was shown to demonstrate an agonistic activity for the cAMP-dependent branch of the melatoninergic pathway [47]. An increased cAMP level and activation of PKA influenced the Kir channels and decreases their activities in the cardiomyocytes [48]. Also, luzindole and 4P-PDOT competitively block MT1 melatonin receptors, and both are inverse agonists in systems with constitutively active MT1 receptors [10, 49-52]. These luzindole effects could lead to a reduction in the amplitude of the IK1 current shown in this study.

Also, we extended the limitation part:

Lines 337-341:

(III) The observed effect of the melatonin receptor blocker luzindole alone might be caused by the pre-existing activity of the melatoninergic pathways in the myocardial tissues. However, this kind of activity could not be characterized in our study, and some mechanistical issues concerning the melatonin/luzindole action remain unresolved.

Comment 3:

The study mentioned DOR for multiple times. However, the definition of DOR was not disclosed. The impact of DOR should also be more offered.

Response:

We included the definition for DOR (dispersion of repolarization) in the appropriate place. Also, a short description of the role of DOR is now also included.

Lines 406-410:

Dispersion of repolarization (DOR), which is generally known to be a prerequisite of the unidirectional conduction block and therefore of reentrant ventricular tachyarrhythmias, was calculated as the difference between the maximal and minimal RTs throughout all leads in the same cardiac cycle in the same recording.

Comment 4:

The format of this article was to place the Method section between Discussion and Conclusion. However, there were several abbreviated terms was used in the Results section, while the definition was in the Method section. Please define the abbreviated terms when they were firstly appeared in the whole article.

Response:

Thank you for the review and notices! We corrected this drawback in the present version.
